# Knee Extension Strength Measures Indicating Probable Sarcopenia Is Associated with Health-Related Outcomes and a Strong Predictor of 1-Year Mortality in Patients Following Hip Fracture Surgery

**DOI:** 10.3390/geriatrics6010008

**Published:** 2021-01-15

**Authors:** Morten Tange Kristensen, Signe Hulsbæk, Louise Lohmann Faber, Lise Kronborg

**Affiliations:** 1Physical Medicine and Rehabilitation Research—Copenhagen (PMR-C), Departments of Physical Therapy and Orthopedic Surgery, Amager-Hvidovre Hospital, University of Copenhagen, 2650 Hvidovre, Denmark; s_hulsbaek@hotmail.com; 2Institute of Clinical Medicine, University of Copenhagen, Nørrebro, 2200 Copenhagen, Denmark; 3Department of Management and Quality, The Greenlandic Health System, Queen Ingrids Hospital, 3900 Nuuk, Greenland, Denmark; louisefaber@gmail.com; 4Department of Midwifery, Physiotherapy, Occupational Therapy and Psychomotor Therapy, Education in Physiotherapy, Faculty of Health, University College Copenhagen, Nørrebro, 2200 Copenhagen, Denmark; lisekronborg@hotmail.com

**Keywords:** hip fractures, sarcopenia, mortality, health-related outcomes, fear of falling, muscle strength

## Abstract

To examine if knee-extension strength (KES) measures indicating probable sarcopenia are associated with health-related outcomes and if KES and hand grip strength (HGS) measures are associated with 1-year mortality after hip fracture. Two groups of older patients with hip fracture had either HGS (*n* = 32) or KES (*n* = 150) assessed during their acute hospital stay. Cut-points for HGS (<27 kg for men and <16 kg for women), and cut-points for maximal isometric KES (non-fractured limb), being the lowest sex-specific quintile (<23.64 kg for men and <15.24 kg for women), were used to examine association with health-related outcomes and 1-year mortality. Overall, 1-year mortality was 12.6% in the two strength groups, of which 47% (HGS) and 46% (KES) respectively, were classified as probable sarcopenia. Probable sarcopenia patients (KES) had lower prefracture function, performed poorly in mobility measures and expressed a greater concern of falling compared to their stronger counterparts. Hazard ratio for 1-year mortality was 2.7 (95%CI = 0.49–14.7, *p* = 0.3) for HGS and 9.8 (95%CI = 2.2–43.0, *p* = 0.002) for KES for probable sarcopenia patients compared to those not. Sex-specific KES measures indicating sarcopenia is associated with health-related outcomes and a strong predictor of 1-year mortality after hip fracture.

## 1. Introduction

The revised European Working Group on Sarcopenia in Older People (EWGSOP2) algorithm for identification of people with sarcopenia has elevated low muscle strength instead of muscle mass as the primary indicator for classifying people with a probable sarcopenia status [1]. and supported by a study of very old men [2]. Hand grip strength (HGS) or the chair stand test are suggested for the assessment of muscle strength in the EWGSOP2, as indicators of a probable sarcopenia status [1]. HGS is considered a proxy measure for the overall strength of older people but a recent study [3], although significant, only found a fair relationship between HGS and knee extension strength (KES) in older women, while a non-significant association between HGS and KES was found in men [4]. The latter study also reported a stronger association between KES measures and performance-based sarcopenia assessment tests (e.g., gait speed) compared to HGS [4]. Correspondingly, Yeung et al. [5] reported a stronger association between health characteristics and KES compared to HGS-measures in community-dwelling older adults referred to a geriatric outpatient clinic, and suggested that KES-measures be part of the comprehensive geriatric assessment. In support, Menant et al. [6] reported that a simple KES assessment in community-dwelling older adults was at least as effective in predicting gait speed, timed up and go and balance performances, when compared to more time-consuming muscle-mass based measures. Menant et al. [6] based their findings on KES cut-points, being the lowest sex-specific quintile; <23.64 kg for men and <15.24 kg for women, for the 419 community-living older people (mean (SD) age of 81.2 (45), years, 49% women) included in their study. The value of applying these KES cut points for patients with hip fracture has not yet been examined. To do this, it is considered important to further validate the predictive value of cut-points used for identifying people with a probable sarcopenic status, and especially for the ultimate endpoint, mortality. Thus, there is no true consensus definition of sarcopenia why an editorial by Mayhew and Raina [7] suggested that “the best definition of sarcopenia will be the one with the strongest association with the health outcomes relevant to sarcopenia”. The chair stand test is considered as alternative to KES measures for the assessment of lower limb strength, which again was suggested as an alternative to HGS in the EWGSOP2 [1]. However, the chair stand test has not proven feasible when used early after hip fracture surgery, due to an almost 100% floor effect [8]. Correspondingly, the proposed EWGSOP2 physical performance measures for identifying severe sarcopenia cases being; gait speed, short physical performance battery, timed up and go and the 400 m walking test, will, when used early after hip fracture, most likely misclassify too many patients as sarcopenic. Thus, walking ability at this time point after hip fracture is highly affected due to the acute trauma, and in accordance with a Korean study [9].

The primary aim of the present study was therefore to examine the association of KES measures indicating probable sarcopenia, using cut-points by Menant et al. [6], with 1-year mortality after hip fracture, and the corresponding associations for HGS in a smaller sample, using cut-points for HGS from the EWGSOP2 [1]. Secondary to investigate the association of KES measures indicating probable sarcopenia with outcomes of physical performance and patient reported measures.

## 2. Materials and Methods

### 2.1. Study Design and Setting

This is a prospective observational study of 182 pre-fracture ambulatory patients with acute hip fracture, of whom 150 had their maximal isometric KES assessed upon acute hospital discharge [10,11,12], and additionally, a group of 32 patients, who had their HGS assessed, were examined in the present study. All patients followed a multidisciplinary enhanced recovery program during their acute care hospital stay in the same specialized hip fracture unit, at Hvidovre University Hospital, Copenhagen, Denmark, from December 2006 to May 2015 (please see Figure 1 for the 4 include groups). The Hospital covers a catchment area of about half a million people and treats about 500 patients with hip fracture a year. The enhanced program included surgery on the day of or day after admission, transfusion if hemoglobin was below 6.0 mmol/L, supplementary oxygen at 2 L/min whenever the patient was supine, and analgesia with epidural from fracture confirmed by X-ray and until the 4th postoperative day [13]. The studies were approved by the local or Capital Region’s Research Ethics Committee (7 December 2007; H-A-2007-0127 + 16 September 2010; 37036). The study applies with the Declaration of Helsinki, all patients gave written informed consent and data collection was registered with the Danish Data Protection Agency (J.nr. 2007-41-1573).

### 2.2. Primary Exposures

Maximal isometric KES at a knee joint angle of 75 degrees was measured in Newtons (converted to kilograms) using a custom-made strain-gauge dynamometer [10] and at a knee joint angle of 90 degrees using a strap-fixated handheld dynamometer [11,12]. The strongest of five and four KES assessments, respectively for each leg was used as datapoints. Cut-point values as defined by Menant et al. [6]; being the lowest sex specific quintile of KES <23.64 kg for men and <15.24 kg for women were applied to the strength of the non-fractured leg to indicate a “probable sarcopenic” status. The non-fractured leg was chosen due to the huge strength loss of more than 50% (% of non-fractured) in the fractured leg early after hip fracture [10]. KES strength measures upon hospital discharge were used while earlier hospital assessments were used in four patients without these data. HGS was measured using an electronic Jamar hand-held dynamometer and a standardized test protocol with the patient positioned in a standard chair with armrests. The strongest of a minimum of three measures [2] with the dominant hand was used as datapoints for HGS. Cut-point values defined by the EWGSOP2(<27 kg for men and <16 kg for women) were applied to HGS measures to indicate a “probable sarcopenic” status [1]. These cut-points are identical with those proposed by the Foundation for the National Institutes of Health Sarcopenia Project (<26 kg for men and <16 kg for women) for identifying women with muscle weakness [14].

### 2.3. Adjustment Variables

The age, sex, pre-fracture physical function using the reliable [15] and modified [16] New Mobility Score (NMS, 0–9 points) [17], and the health status of patients (American Society of Anesthesiologists grade, 1–4 points) [18,19] were chosen as adjustment variables, based on their previously shown influence to mortality after hip fracture [20]. The fracture type (cervical femoral fracture versus trochanteric fracture) was also included in analyses to evaluate the potential influence on mortality, due to its known influence on fracture-related pain, early mobility recovery and muscle strength loss following hip fracture [10,21]. 

### 2.4. Physical Performance and Patient-Reported Measures

Patients with KES measures were evaluated with the performance based Cumulated Ambulation Score [22], Timed Up and Go Test (TUG) [23,24], 10 m gait speed [25], and the questionnaire based Short Falls Efficacy Scale–I (FES-I) [26] upon hospital discharge. Additionally, the place of falling was recorded for the KES group, while the pre-fracture NMS was assessed in both groups. Instructions for the TUG and gait speed testing of patients were “to walk as fast as safely possible” and all patients used a standardized rollator in assessments [27]. All assessments were conducted strictly following standardized written instructions. Performance based measures of KES, HGS, TUG, 10MWT were assessed by three of the authors, while the CAS was assessed by clinicians not otherwise involved. The questionnaire-based FES-I was reported by patients.

### 2.5. Primary and Secondary Outcomes

The date of death within the year following the index hip fracture surgery, was obtained from the Danish Civil Registration system and applied in analyses as post-surgery days to death. Physical performance and patient-reported measures were evaluated for their ability to reflect differences of patients classified as probable sarcopenic versus not and related to their muscle strength assessments in kilograms.

### 2.6. Statistical Analysis

Descriptive statistics were calculated for probable sarcopenic and non-sarcopenic patients and based on their distribution (Q-Q plots), using Student’s *t*-test for continuous data, while chi-square tests were used for categorical data. Corresponding, analysis was used to evaluate the association with health-related outcomes. The association of probable sarcopenia with 1-year post-surgery mortality was evaluated with cox proportional hazards models with 95% confidence intervals (enter method) in crude and multivariable models, and further illustrated with Kaplan–Meier survival graphs for all patients and for the HGS and KES group separately. Log-minus-log versus log of survival time graphs in the Cox models of KES showed parallel curves for the non-sarcopenic (mean days = 361, 95%CI; 353–368) and probable sarcopenic group (mean days = 301, 95%CI; 273–349), *p* < 0.001. A corresponding Hosmer and Lemeshow Test showed that the model adequately seems to fit the data (Chi-square = 7.9, *p* = 0.441). A post-hoc power analysis of mortality for the KES group revealed that a minimum of 44 patients were needed in each group to reach a power of 0.80 with an alpha level of 0.05. Data re reported as mean (SD) or as numbers (percentage). SPSS version 25.0 (IBM Corp, Armonk, BY, USA) was used for the statistical analyses; GraphPad Prism 6.0 software (San Diego, CA, USA) for graphics.

## 3. Results

Characteristics of all patients with KES (*n* = 150) and HGS (*n* = 32) measures respectively, with a mean (SD, range) age of 78.7 (7.8, 60–97) years, admitted from their own home and discharged from the acute hip fracture unit after a mean of 12.3 (6.9) post-surgery days, are shown in Table 1. Seventy-two percent of patients had a high pre-fracture functional level as indicated by a New Mobility Score ≥ 7 points. Besides the KES group being slightly older but with a higher prefracture functional level than the HGS group (*p* < 0.05), no significant differences were seen between the two groups (data not shown). Men were stronger than women in their non-fractured KES measures; mean of 26.3 (9.5) kg (Nm/kg = 1.35 (0.47), Nm = 100.2 (39.1)) versus 17.0 (6.9) kg (Nm/kg = 0.94 (0.34), Nm = 57.2 (22.9)), respectively (*p* < 0.001) and correspondingly in HGS; mean of 29.8 (8.7) kg for men versus 16.8 (4.9) kg for women (*p* = 0.001). The applied cut-points from Menant et al. and the EWGSOP2 classified 46% (KES) and 47% (HGS) of the patients respectively, as probable sarcopenic, and with no significant difference for sex and fracture types within the two groups (Table 1). Patients classified as probable sarcopenic stayed in the acute ward for a mean of 13.7 (7.2) versus 11.2 (6.5) post-surgery days for those not (*p* = 0.02).

### 3.1. KES and HGS-Based Probable Sarcopenia Definitions and 1-Year Death

Overall, 23 out of the 182 (12.6%) patients died within the 1-year post-surgery follow-up which corresponds to 22.6% for those classified as probable sarcopenic and 4.1% if not (HR = 6.3, 95%CI, 2.1–18.4, *p* = 0.001). Correspondingly, HR´s for 1-year mortality was 2.7 (95%CI = 0.49–14.7, *p* = 0.3) for HGS and 9.8 (95%CI = 2.2-43.0, *p* = 0.002) for KES for patients with probable sarcopenia compared to those without. Survival curves are presented for the entire group and the KES and HGS groups, separately (Figure 2A–C, Table 1). Differences were visible in all survival graphs, however the survival curves were only significantly different for the entire group (Figure 2A) and the KES group (Figure 2B), as evaluated by The Mantel–Cox log-rank survival distribution test: x^2^(1) = 15, *p* < 0.001 (both), while this was not the case for the HGS group (Figure 2C): x^2^(1) = 1.4, *p* = 0.2.

The risk of death among the 150 patients with KES measures was significantly increased for men (HR = 3.1), those with a probable sarcopenic status (HR = 7.2) and a low health status (HR = 3.7), while age, BMI and fracture type was not significantly associated with death in multivariable cox regression analysis (Table 2).

### 3.2. KES-Based Probable Sarcopenia Definition and Health-Related Outcomes

The patients probable sarcopenic as indicated by the KES measures performed significantly worse on the 3 health-related performance measures and they expressed a greater fear of falling than their non-sarcopenic counterparts (Table 3, Figure 3). Correspondingly, a significantly lower pre-fracture functional level was reported by patients probable sarcopenic, including those with low HGS measures (pre-fracture New Mobility Score was 6.8 (2.3) for probable sarcopenia patients versus 8.3 (1.5), *p* < 0.001 for not), while patients’ probable sarcopenia based on KES measures were more likely to fall at home (Table 1).

## 4. Discussion

In our sample of patients with hip fracture, sex-specific KES-based cut-points suggested by Menant et al. [6], was a strong predictor of 1-year mortality for those classified as probable sarcopenic compared to their non-sarcopenic counterparts. Additionally, men and those with a low health status experienced a greater risk of 1-year death. Correspondingly, those probable sarcopenic performed worse in basic mobility, gait speed and timed up and go performances, they reported a greater fear of falling, had a lower pre-fracture functional level and more of them had an indoor fall when fracturing their hip. The HGS-based definition of probable sarcopenia suggested by EWGSOP2 [1], was significantly associated with the pre-fracture functional level, but not significantly associated with the 1-year mortality. The latter probably related to the low sample in the HGS group.

Overall, 46% of patients were classified as probable sarcopenic, according to defined cut-points and this proportion was similar for patients assessed by KES and HGS measures respectively. In slight conflict, but not significant, more women were classified as probable sarcopenic with KES measures while the opposite was the case for HGS. A Korean study with sarcopenia definition based on a combination of HGS (Asian version of HGS cut-points; men <26 kg and women <18 kg) and muscle mass assessments, reported slightly (non-significant) more women as sarcopenic compared to men [9]. Still, results of the present study should be interpreted with caution, and especially for HGS, due to the small sample of men (*n* = 10) evaluated in the HGS group. In comparison, a Taiwan study reported about 50% of their patients with hip fracture as sarcopenic, but with about 20% more men compared to women [28]. A Norwegian study by Steihaug et al. [29], with similar inclusion criterion as the present study, and using the New Mobility Score for identifying patients with an independent pre-fracture walking status, reported 52% of patients with low HGS. Although, based on the former EWGSOP cut-points for HGS (<30 kg for men and <20 kg for women) the number of patients classified as probable sarcopenic by Steihaug et al. [29] are similar to those of the present study, adding external validity to our findings. In comparison Menant et al. [6], evaluating community-dwelling older people without fractures reported only 20% as probable sarcopenic. This indicates that older persons experiencing a fall related hip fracture, already at that time point, seems to have lost important lower limb muscle strength. A strength loss, that in the first place, might have contributed to the fall.

### 4.1. Mortality

Previous studies have reported that age, sex, pre-fracture function and health status [20,30,31], in addition to the post-surgery ambulatory status [32,33,34] are strong predictors of mortality after hip fracture. Men and patients with a low health status in the present study also experienced a more than 3 times (both groups) increased risk (odds ratio ≥ 3.1, *p* ≥ 0.01) of death at one year in adjusted analysis. Correspondingly, Steihaug et al. [29] reported and increased risk (odds ratio 3.6, *p* = 0.02) of death or becoming permanent nursing home resident (combined end-point) 1 year after hip fracture for patients classified as sarcopenic versus not. However, to our knowledge, and although shown in older people without a hip fracture [35,36,37], the present study is the first to show lower limb muscle strength, classified as probable sarcopenic by sex-specific KES measures, as an equally important predictor (odds ratio 7.2, *p* = 0.01) of 1-year mortality following a hip fracture.

This further highlight the importance of older people in general engaging in activities reducing falls and preserving their lower limb muscle strength, and especially following a hip fracture. Positively, muscle strength is considered a modifiable factor as compared to the sex and health status of patients at time of a fragility fractur, and progressive strength training can be commenced in the acute care setting after a hip fracture [11,12] or at later time points [38,39,40]. Importantly, strength training has proven the most important exercise modality for improvement of mobility after hip fracture surgery [41], while exercise interventions that include balance and functional training has proven to reduce falls in older people [42].

### 4.2. Health-Related Outcomes

The study by Menant et al. [6], that served as inspiration for the present study, reported that older people with low KES measures (20% classified as sarcopenic) performed significantly worse in balance, gait speed and TUG tests, and more likely to fall at home within the year following these assessments. Still, differences for, e.g., the TUG (10.8 vs 9.4 s) and gait speed (0.63 vs. 0.69 m/s) were small [6] as compared to 29.9 vs. 19.3 s for the TUG and 0.42 vs. 0.64 m/s for gait speed, respectively, for patients identified as probable sarcopenic versus not in the present study. Further, patients classified as probable sarcopenic in the present study also more often fell at home. In support of KES measures compared to HGS, Yeung et al. [5] also reported a stronger association of sex-specific KES measures with health-related outcomes, including the TUG and SPPB. Of importance, score-differences for the TUG and gait speed of patients identified as probable sarcopenic versus not, largely exceeds the measurement error and what is considered a minimally clinically important difference for these tests in patients with hip fracture [24,43]. Regarding, fear of falling which seems persistent as long as 1 year after a hip fracture [44], this also was higher for the probably sarcopenia group in the present study.

Further, patients in the present study presented KES measures, considerably lower than healthy elderly [45], and that according to cut-points suggested by Manini et al. [46], experience a moderate to high risk, respectively, of future mobility limitations.

### 4.3. Limitations

Our study has some limitations: HGS was assessed in a separate group of 32 patients and KES measures was not available for this group. Correspondingly, the 150 patients with KES measures did not have their HGS measured. Baseline characteristics of the two groups were comparable, but comparison of the predictive value of the two strength measures should not be made based on data from the present study, especially due to the lower number of patients in the HGS group. Further, a recent Spanish study of patients with hip fracture reported HGS as an independent predictor of 1-year mortality, although based on lower cut-points (<23 kg for men; <13 kg for women) [47], than suggested by the EWGSOP2. In comparison a Swedish study of very old men reported that probable sarcopenia (EWGSOP2 definition of HGS) was associated with increased mortality (hazard ratio, 3.26, *p* = 0.007) within a 3-year follow-up period [2]. Thus, HGS seems to matter, although a direct comparison cannot be made with the present study. Furthermore, the overall 1-year mortality rates of 12.6% in the present study are considerably lower than otherwise reported for patients with hip fracture in a consecutive sample. In comparison, national data from Poland show 1-year mortality rates above and just below 30% for men and women, respectively [48]. Lower mortality rates were expected in the present study, as all patients were admitted from their own home, and with an independent prefracture indoor walking ability. Accordingly, if a consecutive sample of older patients with hip fracture—including patients from nursing homes and those with a lower cognitive level—was evaluated, a much larger proportion of patient’s would most likely have been classified as probable sarcopenic. Still, even with this higher functioning group of patients evaluated in the present study, about 50% of patients were classified as probable sarcopenic and negatively associated with health-related outcomes and a highly increased risk of death.

### 4.4. Conclusions

Sex-specific KES measures indicating sarcopenia are associated with health-related outcomes and seems a strong predictor of 1-year mortality after hip fracture. We suggest that improving lower limb muscle strength of these frail patients should be given a high priority during their early cross-continuum rehabilitation program and continued thereafter for their remaining lifetime.

## Figures and Tables

**Figure 1 geriatrics-06-00008-f001:**
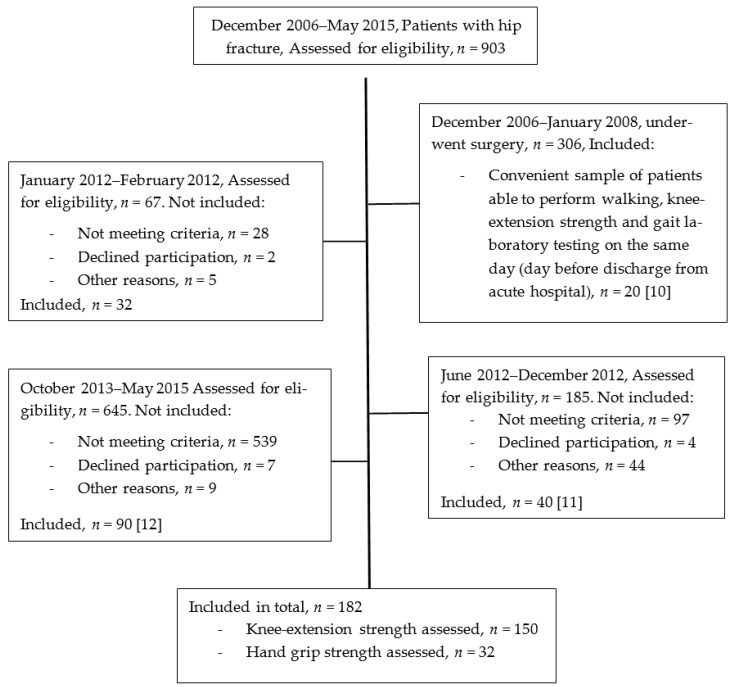
Flowchart of participants, *n* = 182.

**Figure 2 geriatrics-06-00008-f002:**
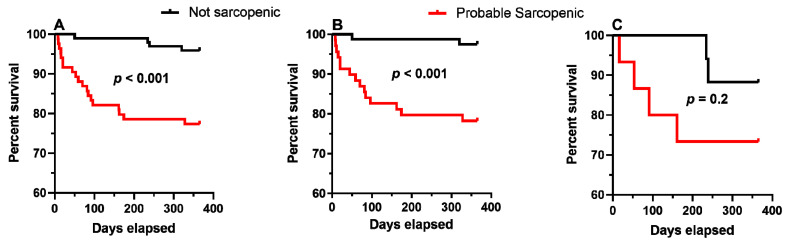
One-year survival curves of patients with hip fracture and a probable sarcopenic status according to muscle strength measures; (**A**) all patients, (**B**) knee-extension strength, (**C**) hand grip strength.

**Figure 3 geriatrics-06-00008-f003:**
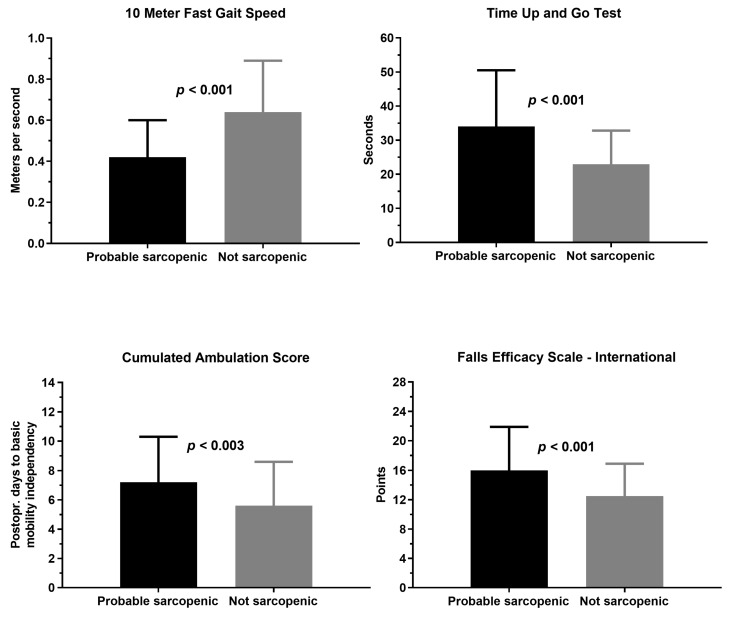
Association of a probable sarcopenic status according to lower limb strength measures and health-related outcomes. Knee extension strength group, *n* = 150. Boxes and whiskers are mean, SD.

**Table 1 geriatrics-06-00008-t001:** Characteristics of patients with hip fracture according to sex-specific muscle strength cut-points of a probable sarcopenic status, *n* = 182.

		Knee-Extension Strength, *n* = 150 Probable Sarcopenic (Menant et al.) [6]			Hand Grip Strength, *n* = 32 Probable Sarcopenic (EWGSOP2) [1]		
Variables	Total, *n* = 182	No, *n* = 81 (54)	Yes, *n* = 69 (46)	*p*	No, *n* = 17(53)	Yes, *n* = 15 (47)	*p*
Women	136 (74.7)	59 (52)	55 (48)	0.3	17 (54)	15 (46)	0.8
Men	46 (25.3)	22 (61)	14 (39)		5 (50)	5 (50)	
Age, years, mean (SD) *	78.7 (7.8)	76.6 (7.2)	82.4 (7.0)	<0.001	75.2 (7.8)	77.1 (9.3)	0.6
Weight, kg, mean (SD)	64.1 (13.3)	66.7 (12.8)	61.1 (13.9)	0.01	65.8 (13.1)	61.4 (11.5)	0.3
Height, cm, mean (SD)	166.5 (9.0)	167.7 (8.7)	164.0 (8.4)	0.009	169.2 (10.7)	168.1 (9.7)	0.8
BMI, mean (SD)	23.0 (4.0)	23.6 (3.9)	22.6 (4.3)	0.1	22.8 (3.1)	21.7 (3.6)	0.4
Cervical femoral neck fractures	88 (48.4)	36 (53)	32 (47)	0.8	11 (55)	9 (45)	0.8
Per- and Subtrochanteric fractures	94 (51.6)	45 (55)	37 (45)		6 (50)	6 (50)	
Low pre-fracture function, NMS 2–6 *	51 (28.0)	9 (24)	28 (76)	<0.001	4 (29)	10 (71)	0.03
High Pre-fracture function, NMS 7–9 *	131 (72.0)	72 (64)	41 (36)		13 (72)	5 (28)	
Low cognitive status	9 (4.9)	1 (17)	5 (83)	0.1	0 (0)	3 (100)	0.09
High cognitive status	173 (95.1)	80 (56)	64 (44)		17 (59)	12 (41)	
Low health status, ASA grade 3–4	38 (20.9)	11 (33.3)	22 (66.7)	0.007	1 (20)	4 (80)	0.2
High health status, ASA grade 1–2	144 (79.1)	70 (59.8)	47 (40.2)		16 (59)	11 (41)	
At-home fallers	n/a	33 (43)	43 (57)	0.008	n/a		
Outdoor fallers		48 (65)	23 (35)				
Days to surgery from admission, median (25–75% IQR)	1 (1–1)	1 (1–1)	1 (1–1)	0.4	1 (1–1)	1 (0–1)	0.8
1-year mortality	23 (12.6)	2/81 (2.5)	15/69 (21.7)	<0.001	2/17 (11.8)	4/15 (26.7)	0.4

Data are numbers (%) or as indicated. NMS; New Mobility Score (0–9 points), ASA; American Society of Anesthesiologists grade. * Knee extension strength group was older but had a higher prefracture functional level than the hand grip strength group (*p* < 0.05), otherwise *p* > 0.05 between the two strength groups.

**Table 2 geriatrics-06-00008-t002:** Crude and adjusted hazard ratios of 1-year mortality according to presence of a sex-specific probable sarcopenic status, *n* = 150.

Variables	Univariable Analyses				Multivariable Analyses			
		95.0% CI for Exp(B)				95.0% CI for Exp(B)		
	Exp(B)	Lower	Upper	*p*	Exp(B)	Lower	Upper	*p*
Age, per year older	1.078	1.01	1.15	0.03	1.04	0.97	1.12	0.3
Women (reference)								
Men	2.4	0.89	6.2	0.08	3.1	1.2	8.4	0.03
BMI, per point higher	0.98	0.87	1.1	0.8	0.98	0.87	1.1	0.8
High prefracture level, NMS, 7–9 (reference)								
Low prefracture level NMS, 2–6	3.7	1.4	9.6	0.007	1.5	0.54	4.19	0.4
High health status ASA, 1–2 (reference)								
Low health status ASA, 3–4	5.8	2.2	15.3	<0.001	3.7	1.3	10.3	0.01
Cervical femoral fracture (reference)								
Trochanteric fracture	2.1	0.72	5.83	0.2	1.8	0.61	5.3	0.3
Not sarcopenic * (reference)								
Probable sarcopenic *	9.8	2.2	43.0	0.002	7.2	1.6	33.4	0.01

BMI; body mass index, NMS; New Mobility Score, ASA; American Society of Anesthesiologists grade, * Knee extension strength-based probable sarcopenia definition.

**Table 3 geriatrics-06-00008-t003:** Health-related outcomes of patients with hip fracture with a sex-specific probable sarcopenic status according to their knee-extension muscle strength measures.

Variables	Total, *n* = 150	Probable Sarcopenia		Difference Mean (95%CI)	*p*
		No, *n* = 81	Yes, *n* = 69		
POD of independent mobility, CAS = 6, mean (SD)	6.2 (3.1)	5.6 (3.0)	7.2 (3.1)	1.7 (0.6; 2.8)	0.003
Not independent in basic mobility, CAS < 6, *n* (%)	22 (14.7)	3 (13.6)	19 (86.4)	n/a	<0.001
Independent in basic mobility, CAS = 6, *n* (%)	128 (85.3)	78 (60.9)	50 (39.1)		
Timed Up and Go test, seconds, *n* = 122, mean (SD)	27.2 (13.9)	22.9 (9.9)	34.0 (16.5)	11.1 (6.4; 15.9)	<0.001
Timed Up and Go test ≥ 20 s *, *n* (%)	105 (70)	45 (43)	60 (57)	n/a	<0.001
Timed Up and Go test < 20 s, *n* (%)	45 (30)	36 (80)	9 (20)		
Fast Gait speed, seconds, *n* = 121, mean (SD)	23.5 (17.1)	19.3 (13.4)	29.9 (29.9)	10.6 (4.6; 16.6)	0.002
Fast Gait speed, m/s, *n* = 121, mean (SD)	0.55 (0.24)	0.64 (0.25)	0.42 (0.18)	−0.21 (−0.29; −0.13)	<0.001
Gait speed ≤ 0.8 s *, *n* (%)	132 (88)	64 (48)	68 (52)	n/a	<0.001
Fast Gait speed > 0.8 s, *n* (%)	18 (12)	17 (94)	1 (6)		
Short Falls Efficacy Scale–I, *n* = 131, mean (SD)	14.0 (5.4)	12.5 (4.4)	16.0 (5.9)	3.5 (1.7; 5.3)	<0.001
Fractured knee-extension strength, Nm/kg, mean (SD)	0.57 (0.32)	0.70 (0.35)	0.43 (0.20)	−0.27 (−0.36; −0.17)	<0.001
Non-fractured Knee-extension strength, Nm/kg, mean (SD)	1.04 (0.41)	1.31 (0.32)	0.72 (0.24)	−0.60 (−0.69; −0.50)	<0.001
Non-fractured knee-extension strength, kg, mean (SD)	19.2 (8.5)	25.0 (6.9)	12.4 (4.0)	−12.6 (−14.5; −10.8)	<0.001
POD of strength testing, mean (SD)	8.2 (2.8)	7.8 (2.6)	8.7 (2.9)	0.94 (0.1; 1.8)	0.04

CAS; Cumulated Ambulation Score, POD; postoperative day, * patients not able to perform included.

## Data Availability

Data are available from the corresponding author on request.

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
