# Peer review of "Knee Extension Strength Measures Indicating Probable Sarcopenia Is Associated with Health-Related Outcomes and a Strong Predictor of 1-Year Mortality in Patients Following Hip Fracture Surgery"

_geriatrics, 2021, doi:10.3390/geriatrics6010008_

Round 1
Reviewer 1 Report
I enjoyed reviewing the paper by Kristensen et al. entitled "Knee extension strength measures indicating probable sarcopenia is associated with health-related outcomes and a strong predictor of 1-year mortality in patients following hip fracture surgery". In this study, the authors focused on the association among knee-extension strength (KES), hand grip strength (HGS), several health-related outcomes and 1-year mortality. By conducting prospective observational study including 182 patients with hip fracture, they found that low KES and HGS were associated with impaired health-related outcomes and 1-year mortality after hip fracture. From these observations, they claimed the importance of preserving muscle strength in patients who underwent hip fracture. These findings will be of interest to the researchers in the field. However, I have following concerns. I hope these comments help you revising your manuscript.
1.
It is well known that sarcopenia is both associated with impaired survival and functional outcome. I am wondering what this paper adds to the existing literature and knowledge. Does this paper advance the concept and how? Please clarify.
2.
Please present key elements of study design early in the paper (i.e. multicenter prospective observational trial).
3.
Please include the relevant date of ethical approval.
4.
Please describe your study settings more in details (e.g. location, tertiary and referral hospitals, academic or community hospitals, number of hospital beds, and number of annual hip fracture admission, and number of surgeries) where the data were collected. I believe such information would help readers to depict the context of your study more accurately. The authors also need to describe the relevant dates, including periods of recruitment, exposure, follow-up, and data collection.
5.
Who did measure the KES and HGS? Who did evaluate Cumulated Ambulation Score, Timed Up & Go test (TUG), 10-meter gait speed, and the questionnaire based Short Falls Efficacy Scale – I (FES-I)? How did you assure the data validity and quality? As these parameters were main outcomes, please clarify more in details. I think If the same researchers are involved in study planning and outcome measurement, there is a theoretical risk of biased assessment.
6.
Sample size estimation
Power analysis is missing: Please explain how the study size was arrived at.
7.
Participants flow chart is missing. Please consider to report numbers of individuals at each stage of study—eg numbers potentially eligible, examined for eligibility, confirmed eligible, included in the study, completing follow-up, and analyzed using a flow diagram, according to the STROBE Statement.
8
Table 1
Many vital information is missing, such as preinjury ADL, dementia, interval between injury to operation, comorbidities (COPD, diabetes mellitus, cancer, heart failure, kidney failure), initial vital sings, bleeding, operation time, and the amount of blood transfusion. I think such variable strongly affects your measured outcomes. The authors sholud make efforts to address these unmeasured confounders.
9
Table 2
You should confirm the models’ goodness of fit and discrimination ability using the Hosmer-Lemeshow test and the c statistic, respectively. Please revise the methods to incorporate with this change.
10
Figure 2
Please define what the error bars stand for. Standard deviation or Standard Error of Mean?
11
In conclusion section, the authors claimed that “These findings emphasize the importance of older people preserving their muscle strength and that improving lower limb muscle strength of patients following a hip fracture, should be given a high priority during their early cross-continuum rehabilitation program and continued thereafter for their remaining lifetime”. However, in fact, current manuscript did not provide any data and methodology that support rehabilitation program and improved muscle strength actually lead to improved outcomes.
12
In the references section, there are many examples of self citations. I know this work is based on your previous research, but too many. Can you balance these citations?
Author Response
Dear Editor and Reviewers,
We thank you for your positive attention to our manuscript entitled “Knee extension strength measures indicating probable sarcopenia is associated with health-related outcomes and a strong predictor of 1-year mortality in patients following hip fracture surgery” and hope that you will find that we have addressed your comments and suggestions, satisfactorily.
Please find our point to point answers in the following, and with changes made in the revised manuscript, with the word editor function.
Comments and Suggestions for Authors
Reviewer 1.
I enjoyed reviewing the paper by Kristensen et al. entitled "Knee extension strength measures indicating probable sarcopenia is associated with health-related outcomes and a strong predictor of 1-year mortality in patients following hip fracture surgery". In this study, the authors focused on the association among knee-extension strength (KES), hand grip strength (HGS), several health-related outcomes and 1-year mortality. By conducting prospective observational study including 182 patients with hip fracture, they found that low KES and HGS were associated with impaired health-related outcomes and 1-year mortality after hip fracture. From these observations, they claimed the importance of preserving muscle strength in patients who underwent hip fracture. These findings will be of interest to the researchers in the field. However, I have following concerns. I hope these comments help you revising your manuscript.
1.
It is well known that sarcopenia is both associated with impaired survival and functional outcome. I am wondering what this paper adds to the existing literature and knowledge. Does this paper advance the concept and how? Please clarify.
Answer: We fully agree with you regarding the mentioned associations for older people. However, to our knowledge, the importance of knee-extension strength of patients with hip fracture indicating sarcopenia, and measured in an acute care setting, which are the key independent variable of this study, is considered new knowledge. Thus, even as patients included are among the better functioning (72% had a high prefracture functional level, only 5% had a low cognitive stautus, and patients from nursing home were not included), we still identified about 50% of patients with strong indications of sarcopenia and associated with performances and high long-term mortality rates. Since all deaths happened after hospital discharge, identifying high risk patients at time of discharge, by measuring their knee-extension strength, potentially seems relevant for identifying patients that should be followed more intensively by health care professionals in the community, which might save lives.
2.
Please present key elements of study design early in the paper (i.e. multicenter prospective observational trial).
Answer: Please see answer to suggestion 4.
3.
Please include the relevant date of ethical approval.
Answer: Please see answer to suggestion 4.
4.
Please describe your study settings more in details (e.g. location, tertiary and referral hospitals, academic or community hospitals, number of hospital beds, and number of annual hip fracture admission, and number of surgeries) where the data were collected. I believe such information would help readers to depict the context of your study more accurately. The authors also need to describe the relevant dates, including periods of recruitment, exposure, follow-up, and data collection.
Answer, Page 2, Line 71-85:
2.1. Study design and setting
This is a prospective observational study of 182 pre-fracture ambulatory patients with acute hip fracture, of whom 150 had their maximal isometric KES assessed upon acute hospital discharge [10,11,12], and additionally, a group of 32 patients, who had their HGS assessed, were examined in the present study. All patients followed a multidisciplinary enhanced recovery program during their acute care hospital stay in the same specialized hip fracture unit, at Hvidovre University Hospital, Copenhagen, Denmark, from January 2007 to May 2015. The Hospital covers a catchment area of about half a million people and treats about 500 patients with hip fracture a year. The enhanced program included surgery on the day of or day after admission, transfusion if hemoglobin was below 6.0 mmol/L, supplementary oxygen at 2 L/min whenever the patient was supine, and analgesia with epidural from fracture confirmed by x-ray and until the 4th postoperative day. The studies were approved by the local or Capital Region’s Research Ethics Committee (7th December 2007; H-A-2007-0127 + 16 September 2010; 37036). The studies apply with the Declaration of Helsinki, all patients gave written informed consent and data collection was registered with the Danish Data Protection Agency (J.nr. 2007-41-1573).
5.
Who did measure the KES and HGS? Who did evaluate Cumulated Ambulation Score, Timed Up & Go test (TUG), 10-meter gait speed, and the questionnaire based Short Falls Efficacy Scale – I (FES-I)? How did you assure the data validity and quality? As these parameters were main outcomes, please clarify more in details. I think If the same researchers are involved in study planning and outcome measurement, there is a theoretical risk of biased assessment.
Answer: All assessments were conducted strictly following standardized written instructions. Performance based measures of KES, HGS, TUG, 10MWT were assessed by 3 of the authors, while the CAS was assessed by clinicians not otherwise involved. The Short FES-I was reported by patients. Thus, other persons than the researchers took part in data collection, and with similar findings across the different measurements. Also, the focus on sarcopenia and mortality, were not known by the researchers at the time of data collection. Thus, we do not believe that the researchers involved introduced bias to our findings.
The following was added to methods, Page 3, Line 117-120: “All assessments were conducted strictly following standardized written instructions. Performance based measures of KES, HGS, TUG, 10MWT were assessed by 3 of the authors, while the CAS was assessed by clinicians not otherwise involved. The questionnaire-based FES-I was reported by patients”.
6.
Sample size estimation
Power analysis is missing: Please explain how the study size was arrived at.
Answer: No a priori power analysis was made. However the following analysis were made to evaluate the robustness of our statistical models and findings, Page 3-4, Line 134-140:
“Log-minus-log versus log of survival time graphs in the Cox models of KES showed parallel curves for the non-sarcopenic (mean days = 361, 95%CI; 353-368) and probable sarcopenic group (mean days = 301, 95%CI; 273-349), p<0.001. A corresponding Hosmer and Lemeshow Test showed that the model adequately seems to fit the data (Chi-square = 7.9, p = 0.441). A post-hoc power analysis of mortality for the KES group revealed that a minimum of 44 patients were needed in each group to reach a power of 0.80 with an Alpha level of 0.05”.
7.
Participants flow chart is missing. Please consider to report numbers of individuals at each stage of study—eg numbers potentially eligible, examined for eligibility, confirmed eligible, included in the study, completing follow-up, and analyzed using a flow diagram, according to the STROBE Statement.
Answer: As mentioned at Page 2, line 72-75, the present study is based on: “Data of 182 pre-fracture ambulatory patients with acute hip fracture, of whom 150 had their maximal isometric KES assessed upon acute hospital discharge [10,11,12], and additionally, a group of 32 patients, who had their HGS assessed, were examined in the present study”.
Patients included in ref 10 were a convenient sample of high level functioning patients at time of hospital discharge, fulfilling strict inclusion criteria’s (should be able to conduct gait and strength testing, in addition to assessment of postural control in a gait laboratory on the same day) and included from January 2007 to January 2008. Patients included in ref 11 and 12 (from June 2nd, 2012 to May 10th, 2015) are shown in detail in STROBE and CONSORT flowcharts, respectively. The additional sample of 32 patients, admitted from their own home, were consecutively included from January 13th, 2012 to February 16th, 2012.
The only follow-up in the present study was related to mortality, which was described as follows in the manuscript at page 3, line x; “The date of death within the year following the index hip fracture surgery, was obtained from the Danish Civil Registration system and applied in analyses as post-surgery days to death”
Its possible to make a flowchart of the entire sample, but since the two largest groups (ref 11 and 12) of the entire sample already are reported in detail in their respective papers, we suggest that it not be included.
Still, if considered mandatory, we will do our best to reconstruct a flowchart of the two other samples. Please also see our response to suggestion 4.
8
Table 1
Many vital information is missing, such as preinjury ADL, dementia, interval between injury to operation, comorbidities (COPD, diabetes mellitus, cancer, heart failure, kidney failure), initial vital sings, bleeding, operation time, and the amount of blood transfusion. I think such variable strongly affects your measured outcomes. The authors sholud make efforts to address these unmeasured confounders.
Data for prefracture functional level (New Mobility Score), cognitive status, and ASA grade (indicating pre-injury health status) were all reported in Table 1 in the original manuscript.
For the revised manuscript, the interval between hospital admission and operation was calculated and added to table 1. As shown patients underwent surgery at median day 1 (IQR, 1-1) after admission and with no significant difference between patients identified as probable sarcopenic versus not. Page 5, Table 1
Regarding data for operation time, bleeding and amount of blood transfusion these data were not collected. However, all patients followed the same standardized liberal transfusion policy, from admission to discharge, where patients were transfused if the hemoglobin dropped below 6.0 mmol/L and monitored closely during their acute hospital stay. Thus, at the time of data collection for the present study (upon hospital discharge) no patient suffered from e.g. anemia.
Information about the liberal transfusion policy was added to the manuscript at Page 2, line 79-80.
9
Table 2
You should confirm the models’ goodness of fit and discrimination ability using the Hosmer-Lemeshow test and the c statistic, respectively. Please revise the methods to incorporate with this change.
Answer: Please see our response to suggestion 6.
10
Figure 2
Please define what the error bars stand for. Standard deviation or Standard Error of Mean?
Answer: The following was added to figure 2 at page 7. Boxes and whiskers are mean, SD
11
In conclusion section, the authors claimed that “These findings emphasize the importance of older people preserving their muscle strength and that improving lower limb muscle strength of patients following a hip fracture, should be given a high priority during their early cross-continuum rehabilitation program and continued thereafter for their remaining lifetime”. However, in fact, current manuscript did not provide any data and methodology that support rehabilitation program and improved muscle strength actually lead to improved outcomes.
Answer: We agree, and revised the conclusion, page 9, line 284-289:
4.4. Conclusion
“Sex-specific KES measures indicating sarcopenia are associated with health-related outcomes and seems a strong predictor of 1-year mortality after hip fracture. We suggest that improving lower limb muscle strength of these frail patients should be given a high priority during their early cross-continuum rehabilitation program and continued thereafter for their remaining lifetime”.
12
In the references section, there are many examples of self citations. I know this work is based on your previous research, but too many. Can you balance these citations?
Answer: Sorry, you are correct. In the manuscript provided by Geriatrics for the revision it was not possible to use a reference system, but 4 references (18, 22, 24 and 25) are highlighted in the reference list, that potentially can be deleted.
Reviewer 2 Report
This manuscript entitled “Knee extension strength measures indicating probable sarcopenia is associated with health-related outcomes and a strong predictor of 1-year mortality in patients following hip fracture surgery” performed prospective study using elderly patients with acute hip fracture aimed to describe the association of knee-extension strength (KES) with 1-year mortality after hip fracture and with patient reported physical performance. The study is interesting and well designed. The methods are sound and the approach is appropriate. Please check some points as described below.
Discussion:
Line 208: It seems that odds ratio is cited only for the male sex, but the p-value for low health status
Line224-226 (Health-related outcomes):You stated that in the study that served as your inspiration (Menant at al) there were 20% sarcopenic patients. Out of all patients in both groups in your study, there were 46% and 47% of probable sarcopenic patients, respectively. What could be the explanation for this difference?
Round 2
Reviewer 1 Report
Thank you for your effort and time spent on this revision. Your thorough revision has greatly increased the scientific value and readability of this manuscript. However, the reviewer thinks there is still room for improvement. My remaining comments are as follows:
1
Regarding your answer to my comment #7 "Still, if considered mandatory, we will do our best to reconstruct a flowchart of the two other samples. Please also see our response to suggestion 4."
Yes, you need to do your best to reconstruct a flowchart, as using a flow diagram is strongly recommended by the STROBE statement. The reviewer thinks providing the flowchart is much better way for helping reader’s understanding than providing the reference.
2.
Regarding your answer to my comment #8
You did not fully address my comment #8. Again, Muscle wasting occurs systemically in older people and in many diseases, including chronic obstructive pulmonary disorder, cancer, diabetes, renal failure, cardiac failure [1-3]. Since such comorbidities can affect both your measured outcome and exposures, these factors would be important confounders that cannot be ignored. If you can provide such data in table 1, and adjusted for such important confounders, the reviewer feels that the quality of your manuscript can be improved further. Please elaborate on this.
- Cohen S, Nathan JA, Goldberg AL. Muscle wasting in disease: molecular mechanisms and promising therapies. Nat Rev Drug Discov. 2015;14:58-74.
- Fanzani A, Conraads VM, Penna F, et al: Molecular and cellular mechanisms of skeletal muscle atrophy: an update. J Cachexia Sarcopenia Muscle.3:163-179, 2012.
- Schefold JC, Bierbrauer J, Weber-Carstens S: Intensive care unit-acquired weakness (ICUAW) and muscle wasting in critically ill patients with severe sepsis and septic shock. J Cachexia Sarcopenia Muscle.1:147-157, 2010.
Regarding your answer to my comment #12
"In the manuscript provided by Geriatrics for the revision it was not possible to use a reference system, but 4 references (18, 22, 24 and 25) are highlighted in the reference list, that potentially can be deleted"
Yes, to balance and ensure sufficient diversity of references, you need to delete or replace these self-citation references (18, 22, 24 and 25). I cannot understand why it was not possible to use a reference system in this revision.
Round 3
Reviewer 1 Report
I admire your perseverance and enthusiasm on this research project. The revision is satisfactory. Thank you for giving me the opportunity to review this manuscript.